# The Dawn of a New Era in Atopic Dermatitis Treatment

**DOI:** 10.3390/jcm11206145

**Published:** 2022-10-18

**Authors:** Kazuhiko Yamamura, Takeshi Nakahara

**Affiliations:** 1Department of Dermatology, Graduate School of Medical Sciences, Kyushu University, Fukuoka 812-8582, Japan; 2Research and Clinical Center for Yusho and Dioxin, Kyushu University, Fukuoka 812-8582, Japan

**Keywords:** atopic dermatitis, therapy, biologics, small-molecule inhibitors

## Abstract

Atopic dermatitis (AD) is one of the most common chronic inflammatory skin diseases, and the condition is typified by barrier dysfunction and immune dysregulation. Recent studies have characterized various phenotypes and endotypes of AD and elucidated the mechanism. Numerous topical and systemic narrow targeting therapies for AD have been developed according to these findings. Topical medications, including Janus kinase (JAK) inhibitors, phosphodiesterase 4 inhibitors, and the aryl hydrocarbon receptor agonist tapinarof, are effective and safe for AD compared to topical corticosteroids. Oral JAK inhibitors and monoclonal antibodies targeting interleukin (IL)-4, IL-13, IL-31, IL-33, OX40, thymic stromal lymphopoietin, and sphingosine 1-phosphate signaling have displayed outstanding efficacy against moderate-to-severe AD. We are currently in a new era of AD treatment.

## 1. Introduction

Atopic dermatitis (AD) is a chronic, relapsing, inflammatory skin disease characterized by persistent pruritus with barrier dysfunction, microbial dysbiosis, and immune dysregulation [1]. The estimated prevalence of AD is 15–20% in children and 6–10% in adults, among whom 40% are classified as having moderate-to-severe disease [2,3,4,5]. In recent decades, patients have been treated with topical corticosteroids/calcineurin inhibitors, phototherapy, and systemic immunosuppressants. However, many patients require frequent laboratory monitoring during systemic immunosuppressant therapy, and they are undertreated because of concerns regarding adverse effects [6]. Patients with moderate-to-severe AD harbor systemic inflammation/immune abnormalities such as strong Th2 activation, expansion of T cell subsets, and increased levels of pro-inflammatory cytokines, including interleukin (IL)-4, IL-13, and IL-31 [7,8,9,10,11,12,13,14]. Therefore, new agents have been developed to target these cytokines, and they have displayed outstanding efficacy for patients with moderate-to-severe AD. Interestingly, topical phosphodiesterase (PDE) 4 inhibitors and aryl hydrocarbon receptor (AhR) agonists are also effective for AD skin lesions in terms of the restoration of skin barrier function and the regulation of inflammatory cytokine production [15,16,17]. This review discusses the molecular mechanisms and therapeutic targets involved in the pathogenesis of AD.

## 2. Emerging Systemic/Topical Agents

In the past few years, numerous systemic (Table 1) and topical (Table 2) emerging agents have been developed for the treatment of patients with AD.

## 3. Monoclonal Antibodies for Moderate-to-Severe AD

### 3.1. Targeting Th2 and Th2-Associated Cytokines: IL-4, IL-13, IL-33, Thymic Stromal Lymphopoietin, and OX40

Narrow targeting agents for AD have been developed based on its pathogenesis (Figure 1). Accumulating evidence indicates that AD features multiple abnormalities in terms of epidermal barrier dysfunction, immunologic dysregulation, and microbial dysbiosis (e.g., increased abundance of *Staphylococcus aureus* and loss of commensal bacterial species) [1]. AD is considered a disease of Th2 predominance, and blockade of Th2 signaling is highly effective in treatment [37]. Dupilumab, an emerging narrow targeting agent that blocks both IL-4 and IL-13 signaling, has exhibited significant clinical benefits in patients with AD [18,38]. Skin IL-13 expression is correlated with disease severity in patients with AD [39,40]. Furthermore, recent studies have illustrated that the IL-13-specific antagonists tralokinumab and lebrikizumab have similar effects as dupilumab [19,20]. These data indicate that IL-13 acts as a critical cytokine in moderate-to-severe AD [41]. Traditionally, CD4^+^ helper T cells have been implicated as the source of Th2 cytokines. However, group 2 innate lymphoid cells (ILC2s) recently emerged as important contributors to AD through their production of IL-5 and IL-13 [42]. ILC2s, which belong to the larger ILC family, also include group 1 and group 3 ILCs [43]. At the cell surface, ILC2s express receptors for the cytokines IL-25, IL-33, thymic stromal lymphopoietin (TSLP), IL-2, IL-9, and IL-7 [44,45,46,47]. IL-33, an alarmin belonging to the IL-1 family, is mainly produced by keratinocytes in skin after cell death or in response to various stimuli, such as antigen challenges and scratches [48]. Human ILC2s in steady-state skin respond to IL-33 and IL-25 but not to TSLP [49]. Etokimab, a human monoclonal IgG1 antibody that neutralizes the activity of IL-33, proved efficacious for AD in a phase 2a trial [21,22]. TSLP is highly expressed in the skin of patients with AD, similar to IL-33, and it activates human myeloid dendritic cells to induce an inflammatory Th2 response [50,51]. However, tezepelumab, the monoclonal antibody targeting TSLP, did not provide significant improvements in patients with moderate-to-severe AD compared to the effects of placebo in a phase 2a trial [23]. These results indicate that IL-33 might contribute to AD aggravation by being more closely associated with ILC-mediated IL-13 production than TSLP. Conversely, the ligand for OX40 (OX40L, also known as CD134L and CD252) is primarily induced by TSLP [50,51]. OX40L is mainly expressed on antigen-presenting cells, such as activated B cells, dendritic cells, monocytes, and Langerhans cells [52,53,54,55]. OX40 (CD134), the receptor for OX40L, transiently expresses after antigen recognition [56]. It is predominantly expressed on activated/memory CD4^+^ T cells and Tregs, whereas it displays lower expression on CD8^+^ T, NK, and NKT cells [56]. The OX40–OX40L interaction is crucial for Th2 responses generating memory T cells by promoting the survival of effector T cells after antigen priming [57,58,59,60,61]. The OX40L–OX40 axis is a novel therapeutic target in autoimmune and inflammatory diseases, as it directly targets antigen-specific T cells responsible for clinical phenotypes without causing widespread immunosuppression [52,56]. A recent phase 2a clinical trial demonstrated that GBR 830, a humanized monoclonal antibody against OX40 that inhibits OX40–OX40L binding, induced significant progressive tissue and clinical changes in patients with moderate-to-severe AD [24].

### 3.2. Targeting Pruritus and the Th2-Associated Cytokine IL-31

IL-31, a four-helix bundle cytokine belonging to the IL-6 cytokine family, is preferentially produced by activated Th2 cells [62,63,64]. It transmits signals via a heterodimeric receptor composed of IL-31 receptor A (IL31RA) and oncostatin M receptor [64,65]. These receptors are expressed on various cell types, including cutaneous peripheral neurons and dorsal root ganglia neurons [65,66,67,68,69]. IL-31 is considered a major pruritogen in AD, and serum IL-31 levels are correlated with disease severity [14,70]. The humanized monoclonal antibody nemolizumab, which targets IL-31 receptor alfa, displayed an apparent anti-pruritic effect in a phase 3 clinical trial [25]. Although the precise mechanism of IL-31 production has not been fully elucidated, the transcription factor endothelial PAS domain protein 1 (EPAS1) plays a key role in IL-31 induction in AD skin inflammation [71]. Further, 4-(2-(4-isopropylbenzylidene)hydrazineyl)benzoic acid, as a small-molecule inhibitor, suppresses EPAS1-driven IL-31 induction [72]. In addition, *DOCK8* has been identified as a negative regulator of IL-31 production linked to EPAS1 nuclear translocation [71]. It is well known that homozygous and compound heterozygous mutations in *DOCK8* cause combined immunodeficiency characterized by recurrent viral infections, early-onset malignancy, and AD [73,74,75,76,77]. Consistent with this, *DOCK8* polymorphism is associated with serum IL-31 levels in patients with moderate-to-severe AD [78].

### 3.3. Targeting Th17-Associated Cytokine IL-17

Psoriasis, along with AD, is one of the most common inflammatory skin diseases. While AD has a strong Th2 component associated with IL-4 and IL-13 over-production, psoriasis is largely driven by Th17 T cells and associated IL-17 activation [79]. IL-17 expression is also enhanced in acute lesions in AD skin compared to uninvolved skin [80], and a correlation between the number of Th17 cells in peripheral blood and acute AD severity has been reported [81]. However, secukinumab, the monoclonal antibody targeting IL-17, did not provide significant improvements in patients with moderate-to-severe AD compared to the effects of placebo in a phase 2 trial [26].

## 4. Targeting Immunomodulatory Effects and Sphingosine 1-Phosphate (S1P) Receptors (S1PRs)

S1P, a bioactive lipid mediator, regulates various cell activities, including cell growth, differentiation, apoptosis, migration, inflammation, metabolism, and angiogenesis [82,83,84]. S1P is secreted by red blood cells, endothelial cells, and platelets into the extracellular environment, and it contributes to several cardiovascular, autoimmune, inflammatory, neurological, oncologic, and fibrotic diseases [85]. In patients with AD, it has been reported that serum S1P levels are elevated and associated with severity [86]. Five subtypes of S1PRs (S1PR1–5) have been identified as seven-membrane-spanning proteins, a characteristic feature of G protein-coupled receptors. S1PR1, S1PR2, and S1PR3 are widely expressed in various tissues, including the brain, lungs, spleen, heart, and kidneys [87]. Unlike S1PR1–3, S1PR4 is expressed in the lungs and lymphoid tissues, and S1PR5 expresses in the brain and skin [87]. Igawa et al. reported that the expression of S1PR1 and S1PR2 is increased in impetigo, a common bacterial skin infection mostly caused by *Staphylococcus aureus* [88]. S1PRs are considered therapeutic targets for patients with AD because agents targeting S1PRs have displayed immunomodulatory effects [89]. In addition, a study using mice reported that S1PR3–TRPA1 signaling contributes to the onset of itches in sensory nerves [90]. Currently, the safety and efficacy of systemic treatment with etrasimod, which targets S1PR1, S1PR4, and S1PR5, has been illustrated in patients with moderate-to-severe AD in a phase 2 clinical trial (NCT04162769), opening the door for this compound to enter phase 3 development.

## 5. Small-Molecule Inhibitors

### 5.1. Janus Kinase (JAK) Inhibitors

IL-4, IL-13, IL-31, and TSLP require downstream JAK-signal transducer and activator of transcription (STAT) signaling [91]. The involvement of all four JAK family members (JAK1–3 and TYK) has been observed in AD, mediating downstream inflammation [92,93]. Phosphorylation of JAK following the binding of a cytokine to its cognate receptor induces the phosphorylation and dimerization of STAT proteins [94]. These STAT proteins regulate target genes after translocating to the nucleus [94,95]. JAK inhibitors inhibit the activity of one or more JAKs, thereby interfering with the JAK–STAT signaling pathway (Figure 2). IL-4 and IL-13 induce JAK1 and JAK3, which activate STAT6 [96]. TSLP and IL-31 induce JAK1 and JAK2 expression, which activates STAT5 [91]. The oral JAK inhibitors baricitinib (JAK1/2), abrocitinib (JAK1-selective), and upadacitinib (JAK1-selective) have been approved for the treatment of AD. All three met primary and secondary endpoints across numerous trials in moderate-to-severe AD [94]. Of patients receiving baricitinib at doses of 1, 2, and 4 mg, EASI-75 scores were significantly higher with the 2 and 4 mg dosages (17% and 21%) than placebo (6%) at week 16 in a phase 3 trial (BREEZE-AD2) [27]. Of patients receiving abrocitinib at doses of 100 and 200 mg, EASI-75 scores were significantly higher with both dosages (45% and 61%) than placebo (10%) at week 12 in a phase 3 trial (JADE-MONO2) [28]. Of patients receiving upadacitinib at doses of 15 and 30 mg, EASI-75 scores were significantly higher with both dosages (60% and 73%) than placebo (13%) at week 16 in a phase 3 trial [29]. These results highlight the importance of Th2 signaling in the pathogenesis of AD. In addition, topical JAK inhibitors such as ruxolitinib (JAK1/2) and delgocitinib (a JAK1/2/3 and Tyk2 inhibitor, i.e., pan-JAK) have also been approved. Ruxolitinib, a first-generation small molecule-inhibitor approved by the FDA, was well tolerated and associated with a low frequency of treatment-emergent adverse events in patients with mild-to-moderate AD [31,32]. Delgocitinib, the world’s first approved topical JAK inhibitor, has been studied in Japan, where it was approved for treating AD in adults and children based on long-term efficacy and safety data [33,34,35].

### 5.2. PDE4 Inhibitors

PDE4 is a key regulator of inflammatory cytokine production in AD through the degradation of cyclic adenosine monophosphate [97,98]. PDE4 inhibitors increase the levels of cyclic adenosine monophosphate in patients with AD and thereby reduce the expression of pro-inflammatory cytokines [99]. The systemic PDE4 inhibitor apremilast did not meet its primary endpoint for patients with moderate-to-severe AD in a double-blind, placebo-controlled PoC trial (NCT02087943) [30]. Conversely, the topical agents crisaborole and difamilast were approved for treating AD in adults and children based on long-term efficacy and safety data in phase 3 trials [15,16,36].

## 6. AhR-Modulating Agent

Tapinarof (GSK2894512, previously WBI-1001) is a naturally derived small molecule produced by bacterial symbionts of entomopathogenic nematodes [100]. It directly binds AhR and activates signaling in multiple cell types, including CD4^+^ T cells and keratinocytes [101]. The ligation of tapinarof and AhR improves the expression of skin barrier genes, regulates the expression of Th2 cytokines, and protects against inflammation-associated oxidative damage [101]. A phase 2b trial revealed that topical tapinarof improved both eczema area and severity index and itch numerical rating scale scores in patients with moderate-to-severe AD with largely mild adverse events [17].

## 7. Conclusions

Emerging topical and systemic targeted agents have been developed on the basis of expanding knowledge of the pathogenesis of AD. These specific cytokine/receptor-targeted agents have displayed safety and efficacy. Moreover, upcoming trials will provide additional therapeutic options for patients with AD. These new therapies also raise problems, such as the long-term socioeconomic burden associated with monoclonal antibody treatments. Thus, we need to choose more appropriate treatments, including combinations of existing therapies. We are currently at the dawn of a new era in the treatment of AD.

## Figures and Tables

**Figure 1 jcm-11-06145-f001:**
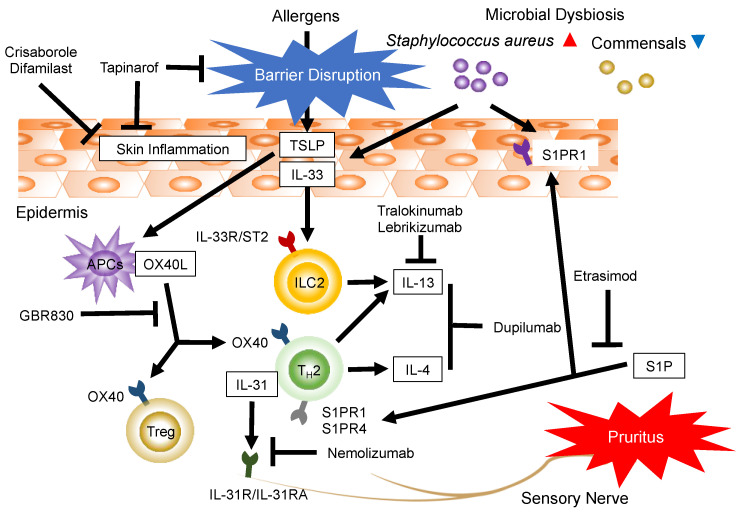
Therapies based on the pathogenesis of atopic dermatitis. Disrupted epidermal barrier function and microbial dysbiosis induce the production of pro-inflammatory mediators. Keratinocyte-produced TSLP and IL-33 enhance type 2 inflammatory responses through the activation of Th2 cells and ILC2s. Th2 cells and ILC2s produce the key inflammatory cytokines (IL-4 and/or IL-13) of AD. The ligation of OX40L and OX40 augments Th2 immune responses. IL-31 is a T cell-derived cytokine associated with pruritus. IL-31 transmits itch sensations via IL-31R in peripheral neurons. The lipid mediator S1P regulates various cell activities, including cell growth, differentiation, apoptosis, migration, inflammation, metabolism, and angiogenesis, through S1PRs. TSLP, thymic stromal lymphopoietin; IL, interleukin; Th, T-helper cells; ILC2s, group 2 innate lymphoid cells; OX40L, OX40 ligand; IL-31R, IL-31 receptor; S1P, sphingosine 1-phosphate.

**Figure 2 jcm-11-06145-f002:**
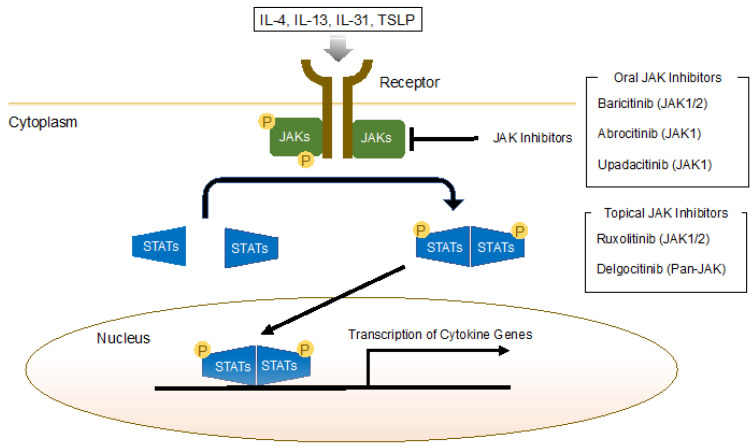
JAK–STAT signaling and oral/topical JAK inhibitors in atopic dermatitis. The JAK protein family (JAK1, JAK2, JAK3, and TYK2) mediates IL-4, IL-13, and IL-31 cytokine signaling via cognate receptors. Activation of JAKs results in the phosphorylation of downstream STAT proteins, followed by their nuclear translocation and activation of target genes. JAK inhibitors inhibit the activity of one or more JAKs, thereby interfering with the JAK–STAT signaling pathway. JAK, Janus kinase; STAT, signal transducer and activator of transcription; TYK2, tyrosine kinase 2; IL, interleukin.

**Table 1 jcm-11-06145-t001:** Emerging Systemic Agents.

	Agent	Target	Study Type	Study Duration	Dose	Reference
Biologics	Dupilumab	IL-4Ra	Phase 3	16 weeks	300 mg	[18]
	Tralokinumab	IL-13	Phase 2b	16 weeks	150, 300 mg	[19]
	Lebrikizumab	IL-13	Phase 2b	16 weeks	125, 250 mg	[20]
	Etokimab	IL-33	Phase 2a	20 weeks	300 mg	[21,22]
	Tezepelumab	TSLP	Phase 2a	12 weeks	280 mg	[23]
	GBR 830	OX40	Phase 2a	16 weeks	10 mg/kg	[24]
	Nemolizumab	IL-31Ra	Phase 3	16 weeks	60 mg	[25]
	Secukinumab	IL-17A	Phase 2	16 weeks	300 mg	[26]
	Etrasimod	S1PR 1/4/5	Phase 2	16 weeks	1, 2 mg	NCT04162769
Small-molecule inhibitors	Baricitinib	JAK 1/2	Phase 3	16 weeks	1, 2, 4 mg	[27]
	Abrocitinib	JAK1	Phase 3	12 weeks	100, 200 mg	[28]
	Upadacitinib	JAK1	Phase 3	16 weeks	15, 30 mg	[29]
	Apremilast	PDE4	Phase 2	12 weeks	30, 40 mg	[30]

**Table 2 jcm-11-06145-t002:** Emerging Topical Agents.

Agent	Target	Study Type	Study Duration	Dose	Reference
Ruxolitinib	JAK1/2	Phase 2	12 weeks	0.15, 0.5, 1.5%	[31,32]
Delgocitinib	Pan-JAK	Phase 3	28 weeks	0.25, 0.5%	[33,34,35]
Crisaborole	PDE4	Phase 3	4 weeks	2.00%	[36]
Difamilast	PDE4	Phase 3	4, 52 weeks	0.3, 1.0%	[15,16]
Tapinarof	AhR	Phase 2b	12 weeks	0.5, 1.0%	[17]

## Data Availability

Not applicable.

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
