# Peer review of "The Dawn of a New Era in Atopic Dermatitis Treatment"

_jcm, 2022, doi:10.3390/jcm11206145_

Round 1

Reviewer 1 Report

The authors present a concise narrative review on the molecular targets of the new therapeutic agents for atopic dermatitis.

The small molecules section is disproportionately smaller than the IL section. Since the review does not really add anything to current knowledge, a discussion on selectivity of Jak inhibition in association with inhibitory concetration the currently available JAKs would be of merit. This a field based on molecular data, and currently on the forefront and  the expert opinion of the authors would be of benefit to the readership

The review is not novel, but it is well-written.

The main question addressed is what are the molecular targets for the new agents for the treatment of AD?

The study does not add anything on our current knowledge.

The conclusions are consistent with the evidence and arguments presented and address the main question posed.

Author Response

The authors present a concise narrative review on the molecular targets of the new therapeutic agents for atopic dermatitis.

The small molecules section is disproportionately smaller than the IL section. Since the review does not really add anything to current knowledge, a discussion on selectivity of Jak inhibition in association with inhibitory concentration the currently available JAKs would be of merit. This a field based on molecular data, and currently on the forefront and the expert opinion of the authors would be of benefit to the readership

Thank you for the insightful suggestion. We have inserted several sentences on selectivity of Jak inhibitors and those of concentration for use of patients with AD in the JAK inhibitors section. Also, we add the tables which summarize systemic/topical agents.

The review is not novel, but it is well-written. The main question addressed is what are the molecular targets for the new agents for the treatment of AD? The study does not add anything on our current knowledge. The conclusions are consistent with the evidence and arguments presented and address the main question posed.

As the reviewer pointed out, the biggest question is what is the most important target for the treatment of AD. Multiple clinical studies have not led to a definitive conclusion, although several cytokines and receptors: IL-4, IL-13, IL-31, IL-33, S1PRs, and AhR play essential roles in the pathogenesis of AD. We believe that upcoming clinical trials will pave the way for that.

Reviewer 2 Report

1. Discussion about advantages and disadvantages of current treatments as well as future approaches (for example, targeting Th17-related cytokines).

2. Tables summarizing the categories, doses, duration of medications for AD in clinical trials should be added.

2. Full names should be mentioned before using abbreviations (for example, TSLP, EASI, NRS).

Author Response

  1. Discussion about advantages and disadvantages of current treatments as well as future approaches (for example, targeting Th17-related cytokines).

We appreciate the reviewer’s insightful comments. Current narrow targeting systemic therapies for AD are effective and safe compared to systemic immunosuppressants. However, long-term socioeconomic burden for monoclonal antibody treatments has become a big problem. We added the term in the conclusion section. Also, we added the section entitled “Targeting Th17-associated cytokine IL-17”.

  1. Tables summarizing the categories, doses, duration of medications for AD in clinical trials should be added.

 We added the two tables summarizing systemic/topical agents according to the reviewer's advice.

  1. Full names should be mentioned before using abbreviations (for example, TSLP, EASI, NRS).

We rechecked and spelled them out in the revised manuscript.

Reviewer 3 Report

This is a fantastic review describing molecular pathogenesis of AD, therapeutics targets and available treatment options. Authors have nicely described systemic as well as topical treatments that has shown efficacy in treating AD. Following see my minor comments,

·       A table summarizing treatments and corresponding publications would be a great addition here.

·       A limitation section would nicely highlight the challenges of current available treatments. For example, long term socioeconomic burden for monoclonal antibody treatments.   

Author Response

This is a fantastic review describing molecular pathogenesis of AD, therapeutics targets and available treatment options. Authors have nicely described systemic as well as topical treatments that has shown efficacy in treating AD. Following see my minor comments,

A table summarizing treatments and corresponding publications would be a great addition here.

Thank you for the very nice suggestion. It is also good for readers. We added the two tables summarizing systemic/topical agents according to the reviewer's advice.

A limitation section would nicely highlight the challenges of current available treatments. For example, long term socioeconomic burden for monoclonal antibody treatments.  

 As the reviewer pointed out, a long-term socioeconomic burden for monoclonal antibody treatments has become a big problem. We inserted the term “These new therapies also raise problems such as long-term socioeconomic burden for monoclonal antibody treatments. Hence, we need to choose more appropriate treatments, including combinations of existing therapies” in the conclusion section.

Reviewer 4 Report

The manuscript is a compact and thorough review discussing the  emerging therapeutic targets and newest treatment of atopic dermatitis (AD). The topic is hot while we're approaching the new era of AD therapy.

Author Response

The manuscript is a compact and thorough review discussing the emerging therapeutic targets and newest treatment of atopic dermatitis (AD). The topic is hot while we're approaching the new era of AD therapy.

Thank you for the nice comments. Although multiple clinical studies have not led to a definitive conclusion, we strongly believe that upcoming clinical trials will pave the way for the best treatment of AD.